# Additional Lymphaticovenular Anastomosis on the Posterior Side for Treatment of Primary Lower Extremity Lymphedema

**DOI:** 10.3390/jcm11030867

**Published:** 2022-02-07

**Authors:** Akitatsu Hayashi, Giuseppe Visconti, Chia-Shen (Johnson) Yang, Nobuko Hayashi, Hidehiko Yoshimatsu

**Affiliations:** 1Department of Lymphedema Center, Kameda General Hospital, Chiba 296-0041, Japan; 2Department of Plastic and Reconstructive Surgery, Università Cattolica del “Sacro Cuore”, University Hospital “A. Gemelli”, Rome 00168, Italy; joevisconti@hotmail.com; 3Division of Plastic and Reconstructive Surgery, Department of Surgery, Kaohsiung Chang Gung Memorial Hospital, Kaohsiung 83301, Taiwan; johnson.c.yang@gmail.com; 4Theory Clinic, Tokyo 104-0061, Japan; nobunobu.komatsu@gmail.com; 5Department of Plastic Surgery, Cancer Institute Hospital of the JFCR, Tokyo 135-0063, Japan; hidehiko.yoshimatsu@gmail.com

**Keywords:** lymphedema, primary lymphedema, lymphaticovenular anastomosis, lymphovenous bypass, supermicrosurgery

## Abstract

The efficacy of lymphaticovenular anastomosis (LVA) for the treatment of primary lymphedema has been reported. Previous research suggested the efficacy of LVA on the anterior side of the lower limb, but no research has yet underlined the effectiveness of LVA on the posterior side. In the present study, we aimed to investigate the efficacy of LVA on the posterior side of the lower leg for treatment of primary lymphedema, i.e., whether further improvement of primary lower extremity lymphedema could be expected by performing LVA on the posterior side of the lower limb in addition to the LVA on the anterior side, which is usually performed. Forty-five patients with primary lower extremity lymphedema who underwent LVA twice between March 2018 and September 2020 were retrospectively investigated. Patients were classified into two groups: those who underwent LVA on the posterior side in the second operation (PoLVA group) and those who underwent LVA on the medial and anterior sides again in the second operation (MeLVA group). All patients underwent LVA on the medial and anterior sides in the first operation, but no sufficient improvement was observed. The following factors in the second operation were compared between the two groups: skin incision length, the number of anastomoses, the diameters of the lymphatic vessels, the time required for the dissection of the lymphatic vessels and veins and the reduction in volume. LVA resulted in 227 anastomoses (106 anastomoses in the PoLVA group and 121 anastomoses in the MeLVA group) in 26 patients with primary lymphedema of the lower extremities in two surgeries. The reduction in lower extremity lymphedema index was significantly greater in the PoLVA group than that in the MeLVA group (10.5 ± 4.5 vs. 5.5 ± 3.6; *p* = 0.008), and the number of anastomoses in the PoLVA group was significantly lower than that in the MeLVA group (3.5 ± 0.6 vs. 4.6 ± 1.0; *p* = 0.038). LVA on the posterior side subsequent to LVA on the medial and anterior sides resulted in the further improvement of primary lower extremity lymphedema with fewer numbers of anastomoses.

## 1. Introduction

Lymphedema is classified into primary and secondary types, depending on the cause. The etiology of secondary lymphedema includes parasitic infection, injury, cancer radiation therapy and lymphadenectomy following gynecologic, urologic and breast cancer [1,2,3,4,5,6,7,8]. Primary lymphedema may be caused by genetic abnormalities, and its prevalence has been reported to be 1 in 6000 to 1 in 10,000 [9]. Some genes have been linked to primary lymphedema, but most cases of primary lymphedema are idiopathic [9,10].

Management options for primary lymphedema include both conservative methods, such as compression therapy, manual lymph drainage and skincare and surgical treatment. Even though a variety of procedures are reported to be effective for the treatment of primary lymphedema, there is not sufficient evidence to determine which surgical procedure is the most appropriate [9,11,12,13]. Among the surgical procedures, lymphaticovenular anastomosis (LVA) is a minimally invasive treatment, requiring only local infiltration anesthesia and small skin incisions [1,14,15,16].

The efficacy of LVA for primary lymphedema has been reported [17,18,19]. Previous research suggested the efficacy of LVA on the medial and anterior sides of lower limbs with patients in the supine position, but no research has yet underlined the effectiveness of LVA on the posterior side with patients in the prone position. We often perform LVA more than once in a patient, including on the posterior side, for primary intractable lymphedema.

In the present study, we aimed to investigate the efficacy of LVA on the posterior side for the treatment of primary lymphedema. We conducted a retrospective observational study to evaluate the effectiveness of this technique and assessed whether LVA on the posterior side subsequent to LVA on the medial and anterior sides resulted in the further improvement of primary lower extremity lymphedema.

## 2. Materials and Methods

We retrospectively investigated patients with primary lower extremity lymphedema who underwent LVA twice between March 2018 and September 2020. The diagnosis of primary lymphedema was made based on the results of indocyanine green (ICG) lymphography and by excluding other edematous diseases [20].

All patients included in this study had received compression therapy using elastic stockings for at least 3 months, which resulted in no clinical improvement. The exclusion criteria included concurrent lymphedema surgical procedures, postoperative compression therapy, less than 6 months of postoperative follow-up and developing lymphedema before 11 years of age. The follow-up period was defined as the time between the first operation and the last outpatient clinic visit after the second operation.

Patients were classified into two groups: those who underwent LVA on the posterior side in the second operation (PoLVA group) and those who underwent LVA on the medial and anterior sides in the second operation (MeLVA group). All patients underwent LVA on the medial and anterior sides in the first operation, but its effect was insufficient. Patients who still complained of edema in the posterior side of the leg after the first operation underwent LVA on the posterior side in the second operation. Those who still had strong complaints of edema on the medial and anterior sides of the leg underwent LVA on the medial and anterior sides again in the second operation. When the first LVA resulted in no improvement of lymphedema, patients underwent combined surgical procedures including LVA, vascularized lymph node transfer and reductive procedure; these patients were excluded from the study.

This study was conducted under the institutional ethical review board. All patients provided written informed consent for participation in this retrospective observational study.

### 2.1. ICG Lymphography

We performed preoperative ICG lymphography for patients who were not allergic to iodine to determine lymphedema severity and to locate the lymphatic vessels [21]. ICG lymphography was performed as follows: 0.2 mL of ICG (Diagnogreen 0.125%; Daiichi Pharmaceutical, Tokyo, Japan) was injected subcutaneously into the first web spaces of the feet and into both edges of the Achilles tendon on the day before surgery. Fluorescent images of lymphatic drainage channels were obtained using a photodynamic eye infrared camera system (PDE; Hamamatsu Photonics K.K., Hamamatsu, Japan) before the operation.

We used lymphedema staging for primary lymphedema described by Hara et al. In their staging system, “no backflow” and “distal backflow” are added to the leg dermal backflow classification described by Yamamoto et al., as the leg dermal backflow stages do not apply to some primary lymphedema patients (Table 1) [19,21].

### 2.2. Lymphaticovenular Anastomosis

LVA was performed using previously documented procedures under local infiltrative anesthesia [22]. When a linear ICG lymphography pattern was observed, approximately 3 cm skin incisions were made on the line. If a linear pattern was not seen, the incisions were made in the region along the great saphenous vein on the medial and anterior sides or along the small saphenous vein on the posterior side, where suitable collecting lymphatic vessels are reported to be present [23]. The great saphenous vein and the small saphenous vein and their branches were preoperatively identified with ultrasound (Figure 1).

Lidocaine (1% or 0.5%) with epinephrine was injected intradermally or subcutaneously in the target region, and skin incisions were made to identify the subcutaneous veins or branches of the small saphenous vein and collecting lymphatic vessels (Figure 2). An elongation of skin incision was performed when the lymphatic vessels or veins could not be found through the initial incision. The veins and the lymphatic vessels were anastomosed in either an end-to-end or end-to-side fashion using 11-0 or 12-0 nylon micro-sutures. The patency of LVA was confirmed by observation of the anastomosis site under a surgical microscope. Preoperative conservative therapy was resumed immediately after the surgery.

### 2.3. Effectiveness of Additional Lymphaticovenular Anastomosis on the Posterior Side

A comparison was made between two groups on the following aspects in the second surgery: patient characteristics (age, body mass index and duration of edema), the number of anastomoses per patient, the diameter of the lymphatic vessels, the required time for dissecting the lymphatic vessels and the veins and skin incision length. Lymphedematous volume was evaluated using a lower extremity lymphedema (LEL) index determined before the second surgery and more than 6 months after the second surgery [24]. The reduction in volume was compared between the two groups.

### 2.4. Statistical Analysis

Analysis was carried out using SPSS (SPSS, Inc., Chicago, IL, USA). All values were reported as mean ± SD. Differences in the means between groups were analyzed by the *t* test. All *p* values were two-sided, and statistical significance was accepted as *p* < 0.05.

## 3. Results

In total, 45 primary lower limb lymphedema patients underwent LVA twice between March 2018 and September 2020. Of the 45 patients, 19 were excluded based on the exclusion criteria (Table 2), whereas the remaining 26 (33 lower limbs) were included in the analysis of the efficacy of LVA.

The characteristics of the included patients (4 men and 22 women) are shown in Table 3. The average age was 44.2 years (range, 16 to 82 years), and the median age of lymphedema onset was 26.3 years (range, 14 to 76 years). The average lymphedema duration was 8.6 years (range, 10 months to 29 years). According to the preoperative leg dermal backflow stage criteria, we classified two limbs as stage 2, seven limbs as stage 3, five limbs as stage 4, seven limbs as stage 5, four limbs as no backflow and eight limbs as distal backflow (Table 3).

Patients were classified into two groups, 13 (1 man and 12 women, 18 limbs) in the PoLVA group and 13 (3 men and 10 women, 15 limbs) in the MeLVA group. The LVA resulted in a total of 227 anastomoses (106 anastomoses in the PoLVA group and 121 anastomoses in the MeLVA group) in 26 patients with primary lymphedema of the lower extremities; the average total number of anastomoses in two surgeries per case was 8.7 (range, 5–12); the average number of anastomoses on the posterior side in the PoLVA group was 3.5 (range, 2–5); the average total operation time of the two surgeries was 406 min (range, 283–725 min); the average total follow-up period was 526 days (range, 205–1089 days); and the average total postoperative reduction in the lower extremity lymphedema index was 18.1 (range, 5.3 to 32.9) (Table 4).

### Effectiveness of Additional Lymphaticovenular Anastomosis on the Posterior Side

The second postoperative volume reduction in the LEL index was significantly greater in the PoLVA group than in the MeLVA group in all subjects (10.5 ± 4.5 vs. 5.5 ± 3.6; *p* = 0.008), in the subject with the first postoperative reduced LEL index score of less than 10 (8.0 ± 3.8 vs. 2.5 ± 1.5; *p* =0.01) and in the subject with the first postoperative reduced LEL index score of more than 10 (13.5 ± 3.5 vs. 7.3 ± 2.8; *p* < 0.01) (Figure 3, Figure 4 and Figure 5). A significant difference was seen between the PoLVA group and the MeLVA group in the number of anastomoses (3.5 ± 0.6 vs. 4.6 ± 1.0; *p* = 0.038) and in skin incision length (3.7 ± 1.1 vs. 2.9 ± 0.6 cm; *p* = 0.042). Incision elongation was necessary in 19 sites out of 46 sites in the PoLVA group, whereas incision elongation was performed in 2 sites out of 59 sites in the MeLVA group. The differences in the diameter of lymphatic vessels and the required time for dissecting lymphatic vessels and veins were not statistically significant between the PoLVA group and the MeLVA group (0.48 ± 0.16 vs. 0.51 ± 0.19 mm; *p* = 0.629, 15.8 ± 2.9 vs. 12.7 ± 2.4 cm; *p* = 0.093) (Table 5).

## 4. Discussion

The present study was conducted to investigate the efficacy of LVA on the posterior side for the treatment of primary lymphedema. The results of this retrospective study demonstrated a statistically significant decrease in postoperative volume in the LEL index and fewer numbers of LVA in the group of those who underwent LVA on the posterior side in the second operation. Thus, LVA on the posterior side is recommended for the second LVA regardless of the results from the first LVA performed on the medial and anterior sides.

Several reports have been published on the efficacy of LVA. O’Brien et al. reported that LVA for primary lymphedema was difficult because the lymphatic vessels were hypoplastic [17]. In contrast, Demirtas et al. stated that LVA was similarly efficacious for both primary and secondary lymphedema [18]. Hara et al. reported that LVA was effective in patients developing primary lymphedema after the age of 11 years [19]. Thus, in this study, we excluded patients whose onset of lymphedema was before the age of 11 years. We often performed LVA more than two times, including not only on the medial and anterior sides but also on the posterior side because all lymphatic drainage routes are likely to be damaged in primary lymphedema. Moreover, we occasionally performed a combined surgical procedure including LVA, vascularized lymph node transfer and a reductive procedure. Even though we excluded patients who underwent a combined procedure in this study, a combination of physiologic and excisional methods was reported to be effective for the treatment in the advanced stages of lymphedema [25,26]. The present retrospective study revealed that the extent of improvement was greater when the second LVA was performed on the posterior side than on the medial and anterior sides.

It has been reported that on the posterior side of the lower extremity, only one or two lymph collecting vessels arise from the lateral side of the Achilles tendon, which travel along the small saphenous vein. Pan et al. demonstrated these vessels drain into the popliteal and the femoral lymph nodes before finally entering the deep inguinal, the external iliac or the inferior gluteal lymph node groups, while most lymphatic vessels in the anterior side drain into the superficial inguinal lymph nodes [23]. Their study revealed that the lymphatic drainage route in the posterior side is completely different from that of the anterior side. This may explain why the decrease in lower extremity lymphedema index was significantly larger in the PoLVA group, making bypasses in completely different lymphatic routes in the second LVA.

In our study, incision elongation was required more often in the PoLVA group, presumably due to the fact that there are fewer lymphatic vessels on the posterior side. For LVA on the posterior side, in addition to an accurate anatomical understanding of the lymphatic routes, a preoperative detection technique of the lymphatic vessels and the branches of the small saphenous vein is also important. ICG lymphography is a minimally invasive imaging modality that can not only evaluate the severity of lymphedema but also determine the location of the lymphatic vessels by visualizing superficial lymph flows [16,21,27,28]. If a linear ICG lymphography pattern is observed, skin incisions can be made on the linear pattern. However, ICG lymphography cannot visualize lymphatic flow in the deep layer of subcutaneous tissue, or one that is masked beneath dermal backflow patterns, particularly in stardust and diffuse patterns [29]. Ultrasonography can detect the lymphatic vessels even in regions with dermal backflow patterns [30,31,32,33]. Ultrasonography provides a roadmap for surgeons when performing LVA. Surgeons may also select the vein with an appropriate size for LVA from among subcutaneous veins and branches of the small saphenous vein on the posterior side by using preoperative ultrasonography on the posterior side. These techniques may aid LVA on the posterior side, prevent intraoperative incision elongation and decrease the operative time.

This study has several limitations. First, we did not statistically examine the relationship between the group that underwent LVA only on the posterior side and the group with LVA only on the medial or anterior side. We usually first perform LVA on the medial and anterior sides for the following reasons: (1) we are familiar with an anatomical understanding of the lymphatic pathways on the medial and anterior sides, and (2) there are more lymphatic vessels on the medial and anterior sides than on the posterior side. Second, the study is limited by its retrospective nature. The effects of LVA include not only volume reduction, but also the reduction in the frequency of cellulitis or softening of the edematous limbs. A prospective study with a large number of patients is needed to take these accounts into consideration.

## 5. Conclusions

To our knowledge, this is the first attempt to investigate the efficacy of LVA on the posterior side of the lower limb for the treatment of primary lymphedema. Although fewer anastomoses were established, LVA on the posterior side subsequent to LVA on the medial and anterior sides resulted in further improvement of primary lower extremity lymphedema with fewer numbers of anastomoses.

## Figures and Tables

**Figure 1 jcm-11-00867-f001:**
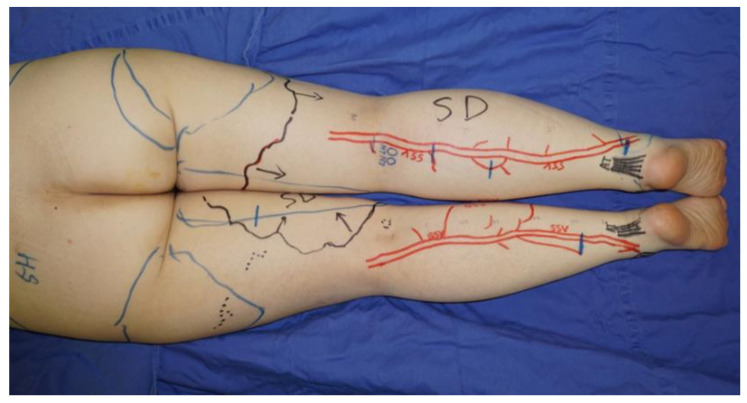
If a linear pattern was not identified with indocyanine green lymphography, the incisions for lymphaticovenular anastomosis were made in the region along the small saphenous vein on the posterior side. The small saphenous vein and their branches were preoperatively identified and marked using ultrasonography.

**Figure 2 jcm-11-00867-f002:**
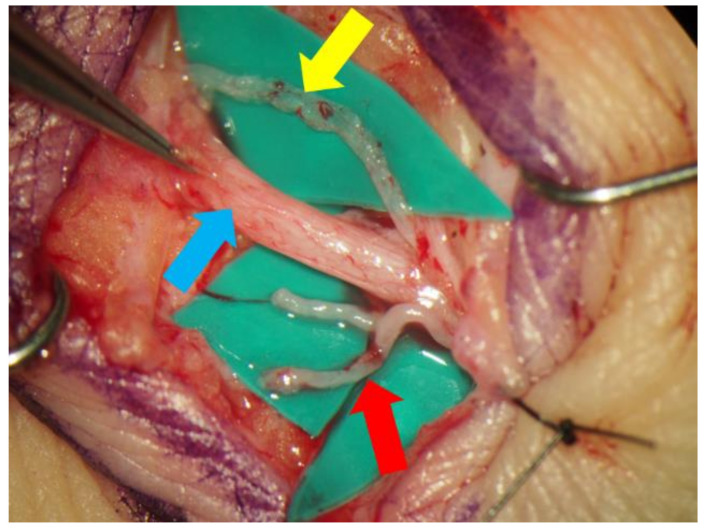
The small saphenous vein (blue arrow) and the lymphatic vessel (yellow arrow) were dissected out via a small incision made in posterior side of the lower leg. In this case, the lymphatic vessel ran along the small saphenous vein. The branches of the saphenous vein (red arrow), which seemed to be suitable for lymphaticovenular anastomosis, were also dissected.

**Figure 3 jcm-11-00867-f003:**
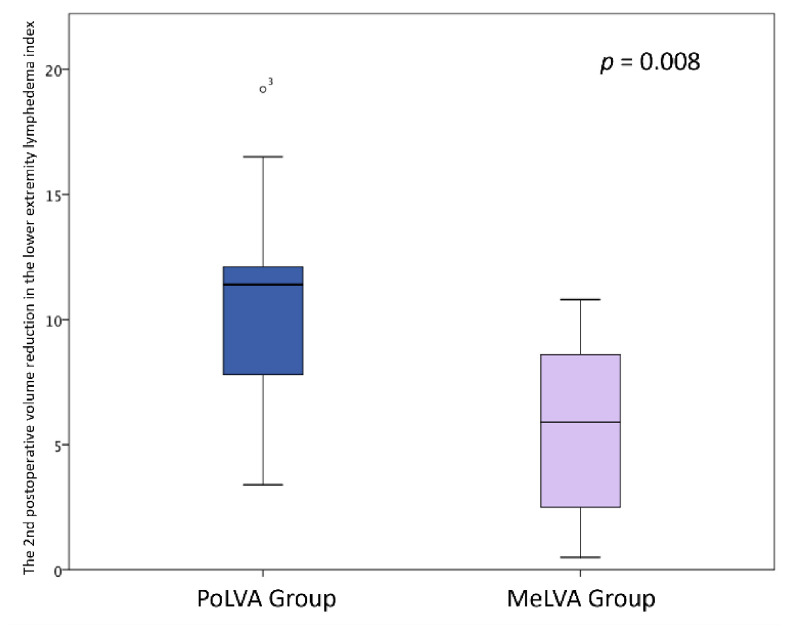
The second postoperative volume reduction in the lower extremity lymphedema index was significantly greater in the PoLVA group than the MeLVA group in all subjects (10.5 ± 4.5 vs. 5.5 ± 3.6; *p* = 0.008). Error bars represent standard error.

**Figure 4 jcm-11-00867-f004:**
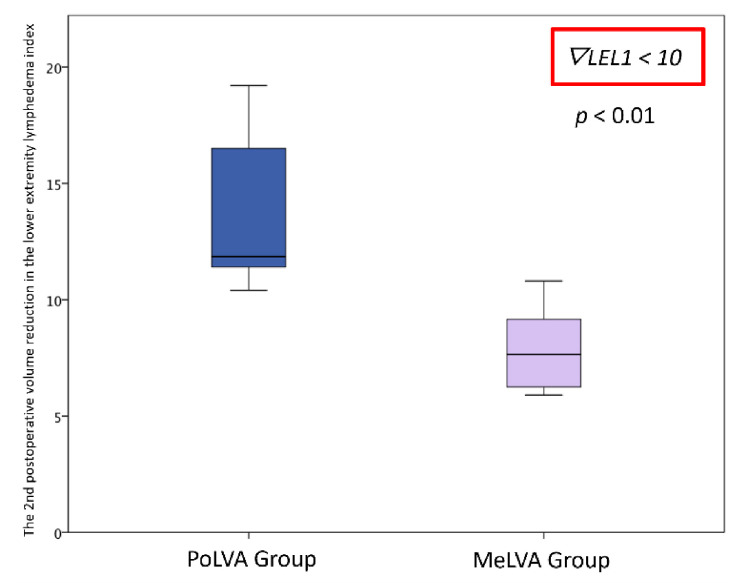
Volume reduction in the lower extremity lymphedema index was significantly greater in the PoLVA group than in the MeLVA group in subject with the first postoperative reduced LEL index score of less than 10 (8.0 ± 3.8 vs. 2.5 ± 1.5; *p* = 0.01). Error bars represent standard error.

**Figure 5 jcm-11-00867-f005:**
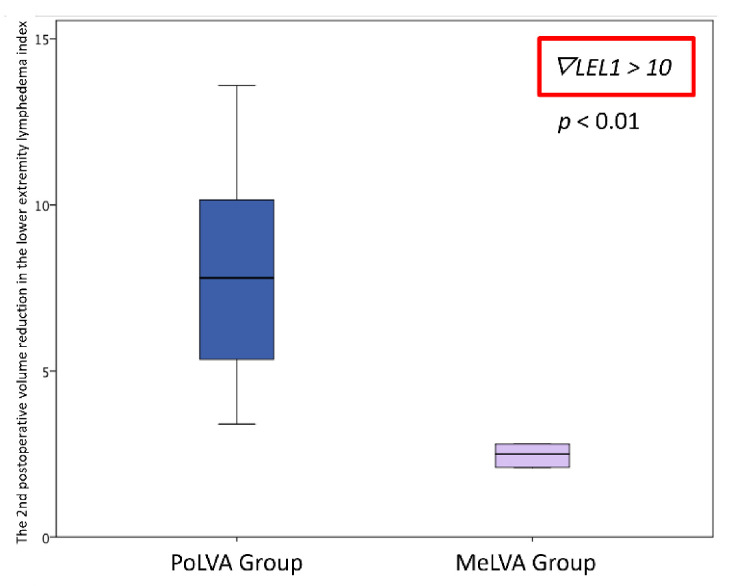
Volume reduction in the lower extremity lymphedema index was significantly greater in the PoLVA group than in the MeLVA group in subject with the first postoperative reduced LEL index score of more than 10 (13.5 ± 3.5 vs. 7.3 ± 2.8; *p* < 0.01). Error bars represent standard error.

**Table 1 jcm-11-00867-t001:** Leg dermal backflow stage chart for primary lymphedema.

Stage	Description
0	No dermal backflow pattern and a clear linear pattern
1	Splash pattern around the groin region
2	Stardust pattern extended from proximal side of lower limb to thigh
3	Stardust pattern extended from proximal side of lower limb to lower leg
4	Stardust pattern extended to the whole limb
5	Existence of a diffuse pattern with stardust pattern in the background
NB	No dermal backflow pattern and some regions without even linear pattern
DB	Stardust pattern or diffuse pattern only in the distal lower limb

NB, no backflow; DB, distal backflow.

**Table 2 jcm-11-00867-t002:** Exclusion criteria for evaluation of the efficacy of lymphaticovenous anastomosis and the number of excluded patients.

Exclusion Criteria	No. of Patients
Including Other Procedure	
-Lymph node transfer	7
-Excision	2
Postoperative stepped-up compression therapy	3
Less than 6 months of postoperative follow-up	5
Developing lymphedema before 11 years of age	2
Total	19

**Table 3 jcm-11-00867-t003:** Patient characteristics.

33 Legs of 26 Lower Limb Primary Lymphedema Patients	
Age, years	16-82 (44.2)
Sex *	
Male	4 (15.4%)
Female	22 (84.6%)
ISL classification *	
1	3 (9.1%)
2a	15 (45.5%)
2b	14 (42.4%)
3	1 (3.0%)
LDB stage *	
Stage 1	0 (0%)
Stage 2	2 (6.1%)
Stage 3	7 (21.2%)
Stage 4	5 (15.2%)
Stage 5	7 (21.2%)
NB	4 (12.1%)
DB	8 (24.2%)
Edema site *	
Bilateral	6 (23.1%)
Right	9 (34.6%)
Left	11 (42.3%)
Duration of edema, years	0.8-29 (8.6)

ISL, International Society of Lymphology; LDB, leg dermal backflow; NB, no backflow; DB, distal backflow; data are ranges (averages) unless otherwise indicated; * Data are counts (percentages).

**Table 4 jcm-11-00867-t004:** Operation summary.

227 LVAs in 26 Primary Lymphedema Patients in Two Surgeries	
No. of total LVAs per patient	5–12 (8.7)
No. of LVAs on posterior side in PoLVA group	2–5 (3.5)
Operative time, min	283–725 (406)
Follow-up period, day	205–1089 (526)
Total postoperative reduction in LEL index	5.3–32.9 (18.1)

LVAs, lymphaticovenular anastomoses; LEL, lower extremity lymphedema. Data are ranges (averages).

**Table 5 jcm-11-00867-t005:** Comparison between cases of PoLVA and MeLVA groups.

	PoLVA Group (*n* = 13)	MeLVA Group (*n* = 13)	*p*
Age, years	42.1 ± 11.7	46.6 ± 13.8	0.586
Body Mass Index, kg/m2	24.2 ± 2.5	23.4 ± 3.2	0.512
Duration of edema, years	9.1 ± 3.8	8.0 ± 3.3	0.316
No. of LVAs in the first operation	4.4 ± 1.5	4.7 ± 1.2	0.487
No. of LVAs in the second operation	3.5 ± 0.6	4.6 ± 1.0	0.038 *
Diameter of lymphatic vessels, mm	0.48 ± 0.16	0.51 ± 0.19	0.629
Required time for dissecting lymphatic vessels and veins, min	15.8 ± 2.9	12.7 ± 2.4	0.093
Length of skin incision, cm	3.7 ± 1.1	2.9 ± 0.6	0.042 *
The first postoperative volume reduction in LEL index	9.3 ± 4.5	10.7 ± 5.0	0.453
The second postoperative volume reduction in LEL index	10.5 ± 4.5	5.5 ± 3.6	0.008 *
Total postoperative volume reduction in LEL index	19.8 ± 8.6	16.2 ± 8.0	0.282

LVAs, lymphaticovenular anastomoses; LEL, lower extremity lymphedema. * *p* < 0.05.

## Data Availability

The data presented in this study are available on request from the corresponding author.

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
