# Peer review of "Additional Lymphaticovenular Anastomosis on the Posterior Side for Treatment of Primary Lower Extremity Lymphedema"

_jcm, 2022, doi:10.3390/jcm11030867_

Round 1
Reviewer 1 Report
The authors present interesting outcomes comparing the efficacy of LVA on the posterior side and medial and anterior side in the second surgery of primary lymphedema cases. They show a statistically significant higher volume reduction with fewer number of LVA and longer skin incision in the posterior LVA group. In the discussion they include a possible explanation of the results obtained and some limitations of their study (the most important, it is a retrospective article).
There are some points I would like to comment to the authors:
- In the second paragraph of the material and methods, it is not clear why there are two definitions of the follow-up period. In the results section, the authors mention only one follow-up. Please explain it, or re-write the description of the follow-up.
- Minor text editing mistakes: "[]" are missed for the reference 21 in the line 91 and reference 17 in the line 228.
- Although the acronym ICG is ver well known, "indocyanine green (ICG)" should be written the first time the authors mention it.
- In the material and methods section, please specify the patient characteristics registered (age, body mass index and duration of edema).
In conclusion, I found this article very valuable, well-planned methodology and it is based on a topic with high potential clinical and research potencial in our time.
Author Response
Dear reviewer,
Thank you for your constructive comments concerning our manuscript “Additional Lymphaticovenular Anastomosis on the Posterior Side for Treatment of Primary Lower Extremity Lymphedema.” We have studied your comments carefully and made corrections, which we hope will meet your approval. We answered your questions or comments in detail in the following paragraphs. The manuscript including the legends of figure was checked by a native English-speaking colleague again. All changes to the manuscript are marked up using the “Track Changes” function.
Author's Response To Reviewer Comments
ITEMIZED, POINT-BY-POINT RESPONSES TO THE COMMENTS OF THE REVIEWERS
Comment 1: In the second paragraph of the material and methods, it is not clear why there are two definitions of the follow-up period. In the results section, the authors mention only one follow-up. Please explain it, or re-write the description of the follow-up.
Response 1: We are sorry for this unclear description. The following sentence was replaced with original sentence in the second paragraph according to the suggestion: “The follow-up period was defined as the time between the first operation and the last outpatient clinic visit after the second operation.”
Comment 2: Minor text editing mistakes: "[]" are missed for the reference 21 in the line 91 and reference 17 in the line 228.
Response 2: We are sorry for this mistake. "[]" were inserted for the reference 21 in the line 94 and reference 17 in the line 236.
Comment 3: Although the acronym ICG is well known, "indocyanine green (ICG)" should be written the first time the authors mention it.
Response 3: Thank you for this comment. "ICG" was replaced with “indocyanine green” in this manuscript after that "indocyanine green (ICG)" was written the first time we mention this according to the suggestion.
Comment 4: In the material and methods section, please specify the patient characteristics registered (age, body mass index and duration of edema).
Response 4: Thank you for the suggestion. The collected patient characteristics were added in the Materials and Methods section according to the suggestion: “Comparison was made between two groups on the following aspects in the second surgery: patient characteristics (age, body mass index and duration of edema),….”
We hope these modifications will obtain your approval. Thank you very much for your consideration of this paper.
Yours sincerely,
Akitatsu Hayashi
January 23, 2022
Reviewer 2 Report
The authors described an interesting retrospective study. 26 patients underwent LVA twice. Patients were divided in two groups: those who underwent LVA on the posterior side in the second operation [PoLVA group] and those who underwent LVA on the medial and anterior side again in the second operation [MeLVA group].
The manuscript has a good structure; however I have some concerns and some recommendations to improve it.
The manuscript will benefit a lot from extensive editing for grammar. (i.e. Legend of figure 2 should be revised)
How did the authors decide whether performing anterior or posterior LVA in the second surgical step?
After the first LVA procedure, when the outcome seemed to be insufficient, how do the authors decide whether performing another LVA, VLNT or reductive procedure?
Combination of physiologic and excisional methods was reported to be effective for the treatment of moderate and advanced stages lymphedema.
Please address and discuss this topic commenting the followings
Campisi CC, et al. Fibro-Lipo-Lymph-Aspiration With a Lymph Vessel Sparing Procedure to Treat Advanced Lymphedema After Multiple Lymphatic-Venous Anastomoses: The Complete Treatment Protocol. Ann Plast Surg. 2017 Feb;78(2):184-190. doi: 10.1097/SAP.0000000000000853
Bolletta A, et al. Combined lymph node transfer and suction-assisted lipectomy in lymphedema treatment: A prospective study. Microsurgery. 2022 Jan 7. doi: 10.1002/micr.30855
Author Response
Dear reviewer,
Thank you for your constructive comments concerning our manuscript “Additional Lymphaticovenular Anastomosis on the Posterior Side for Treatment of Primary Lower Extremity Lymphedema.” We have studied your comments carefully and made corrections, which we hope will meet your approval. We answered your questions or comments in detail in the following paragraphs. All changes to the manuscript are marked up using the “Track Changes” function.
Author's Response To Reviewer Comments
ITEMIZED, POINT-BY-POINT RESPONSES TO THE COMMENTS OF THE REVIEWER
Comment 1: The manuscript will benefit a lot from extensive editing for grammar. (i.e. Legend of figure 2 should be revised)
Response1: Thank you for the suggestion. The manuscript including the legends of figure was checked by a native English-speaking colleague again. (i.e. Legend of figure 2 was revised according to the suggestion: “The small saphenous vein (blue arrow) and the lymphatic vessel (yellow arrow) were dissected out via a small incision made in posterior side of the lower leg. In this case, the lymphatic vessel ran along the small saphenous vein. The branches of the saphenous vein (red arrow), which seemed to be suitable for lymphaticovenular anastomosis, were also dissected.”
Comment 2: How did the authors decide whether performing anterior or posterior LVA in the second surgical step?
Response 2: Thank you for this comment. We are sorry for the lack of explanation. The following sentence was added to the Materials and Methods: “Patients who still complained of edema in the posterior side of the leg after the first operation underwent LVA on the posterior side in the second operation. Those who still had strong complaints of edema on the medial and anterior side of the leg underwent LVA on the medial and anterior side again in the second operation.”
Comment 3: After the first LVA procedure, when the outcome seemed to be insufficient, how do the authors decide whether performing another LVA, VLNT or reductive procedure?
Response 3: Thank you for this comment. We are sorry for the lack of explanation. The following sentence was added to the Materials and Methods: “When the first LVA resulted in no improvement of lymphedema, patients underwent combined surgical procedure including LVA, vascularized lymph node transfer and reductive procedure; these patients were excluded from the study.”
Comment 4: Combination of physiologic and excisional methods was reported to be effective for the treatment of moderate and advanced stages lymphedema. Please address and discuss this topic commenting the followings.
Response 4: Thank you for the suggestion. The following sentence was added to the Discussion: “Moreover, we occasionally perform combined surgical procedure using LVA, vascularized lymph node transfer and reductive procedure. Even though we excluded patients who underwent combined procedure in this study, combination of physiologic and excisional methods was reported to be effective for the treatment in the advanced stages of lymphedema.”, and the below papers were added to References according to the suggestion.
- Campisi CC; Ryan M; Boccardo F; Campisi C. Fibro-Lipo-Lymph-Aspiration With a Lymph Vessel Sparing Procedure to Treat Advanced Lymphedema After Multiple Lymphatic-Venous Anastomoses: The Complete Treatment Protocol. Ann Plast Surg. 2017, 78, 184-190.
- Bolletta A; Taranto G; Losco L; Elia R; Sert G; Ribuffo D; Cigna E; Chen H-C. Combined lymph node transfer and suc-tion-assisted lipectomy in lymphedema treatment: A prospective study. Microsurgery. 2022, Jan 7. doi: 10.1002/micr.30855. Online ahead of print.
We hope these modifications will obtain your approval. Thank you very much for your consideration of this paper.
Yours sincerely,
Akitatsu Hayashi
January 23, 2022
Round 2
Reviewer 2 Report
The authors responded to this reviewer comments.